# Interior-point Methods Strike Back: Solving the Wasserstein Barycenter Problem

**Dongdong Ge**
Research Institute for Interdisciplinary Sciences
Shanghai University of Finance and Economics
ge.dongdong@mail.shufe.edu.cn

**Haoyue Wang***
School of Mathematical Sciences
Fudan University
haoyuewang14@fudan.edu.cn

**Zikai Xiong***
School of Mathematical Sciences
Fudan University
zkxiong16@fudan.edu.cn

**Yinyu Ye**
Department of Management Science and Engineering
Stanford University
yyye@stanford.edu

## Abstract

Computing the Wasserstein barycenter of a set of probability measures under the optimal transport metric can quickly become prohibitive for traditional second-order algorithms, such as interior-point methods, as the support size of the measures increases. In this paper, we overcome the difficulty by developing a new adapted interior-point method that fully exploits the problem's special matrix structure to reduce the iteration complexity and speed up the Newton procedure. Different from regularization approaches, our method achieves a well-balanced tradeoff between accuracy and speed. A numerical comparison on various distributions with existing algorithms exhibits the computational advantages of our approach. Moreover, we demonstrate the practicality of our algorithm on image benchmark problems including MNIST and Fashion-MNIST.

## 1 Introduction

To compare, summarize, and combine probability measures defined on a space is a fundamental task in statistics and machine learning. Given support points of probability measures in a metric space and a transportation cost function (e.g. the Euclidean distance), Wasserstein distance defines a distance between two measures as the minimal transportation cost between them. This notion of distance leads to a host of important applications, including text classification [30], clustering [25, 26, 15, 31], unsupervised learning [23, 13], semi-supervised learning [47], supervised-learning[27, 19], statistics [38, 39, 48, 21], and others [7, 41, 1, 44, 37]. Given a set of measures in the same space, the 2-Wasserstein barycenter is defined as the measure minimizing the sum of squared 2-Wasserstein distances to all measures in the set. For example, if a set of images (with common structure but varying noise) are modeled as probability measures, then the Wasserstein barycenter is a mixture of the images that share this common structure. The Wasserstein barycenter better captures the underlying geometric structure than the barycenter defined by the Euclidean or other distances. As a result, the Wasserstein barycenter has applications in clustering [25, 26, 15], image retrieval [14] and others [32, 43, 11, 34].

From the computation point of view, finding the barycenter of a set of discrete measures can be formulated by linear programming[6, 8]. Nonetheless, state-of-the-art linear programming solvers do not scale with the immense amount of data involved in barycenter calculations. Current research

on computation mainly follows two types of methods. The first type attempts to solve the linear program (or some equivalent problem) with scalable first-order methods. J.Ye et al. [54] use modified Bregman ADMM(BADMM) – introduced by [51] – to compute Wasserestein barycenters for clustering problems. L.Yang et al. [53] adopt symmetric Gauss-Seidel ADMM to solve the dual linear program, which reduces the computational cost in each iteration. S.Claici et al. [12] introduce a stochastic alternating algorithm that can handle continuous input measures. However, these methods are still computationally inefficient when the number of support points of the input measures and the number of input measures are large. Due to the nature of the first-order methods, these algorithms often converge too slowly to reach high-accuracy solutions.

The second, more mainstream, approach introduces an entropy regularization term to the linear programming formulation[14, 9]. This technique was first developed in solving optimal transportation problem. See [14, 3, 18, 50, 33, 10, 22] for the related works. M. Staib et al. [49] discuss the parallel computation issue and introduce a sampling method. P.Dvurechenskii et al. [17] study decentralized and distributed computation for the regularized problem. These methods are indeed suitable for large-scale problems due to their low computational cost and parsimonious memory usage. However, this advantage is obtained at the expense of the solution accuracy: especially when the regularization term is weighted less in order to approximate the original problem more accurately, computational efficiency degenerates and the outputs become unstable [9]. S. Amari et al. [5] propose a entropic regularization based sharpening technique but their result is not the accurate real barycenter. P.C. Alvarez-Esteban et al. [4] prove that the barycenter must be the fixed-point of a new operator. See [42] for a detailed survey of related algorithms.

In this paper, we develop a new interior-point method (IPM), namely Matrix-based Adaptive Alternating interior-point Method (MAAIPM), to efficiently calculate the Wasserstein barycenters. If the support is pre-specified, we apply one step of the Mizuno-Todd-Ye predictor-corrector IPM[36]. The algorithm gains a quadratic convergence rate showed by Y. Ye et al. [56], which is a distinct advantage of IPMs over first-order methods. In practice, we implement Mehrotra's predictor-corrector IPM [35], and add clever heuristics in choosing step lengths and centering parameters. If the support is also to be optimized, MAAIPM alternatively updates support and linear program variables in an adaptive strategy. At the beginning, MAAIPM updates support points $X^*$ by an unconstrained quadratic program after a few number of IPM iterations. At the end, MAAIPM updates $X^*$ after every IPM iteration and applies the "jump" tricks to escape local minima. Under the framework of MAAIPM, we present two block matrix-based accelerated algorithms to quickly solve the Newton equations at each iteration. Despite a prevailing belief that IPMs are inefficient for large-scale cases, we show that such an inefficiency can be overcome through careful manipulation of the block-data structure of the normal equation. As a result, our stylized IPM has the following advantages.

**Low theoretical complexity.** The linear programming formulation of the Wasserstein barycenter has $m \sum_{i=1}^{N} m_i + m$ variables andred$Nm + \sum_{i=1}^{N} m_i + 1$ constraints, where the integers $N$, $m$ and $m_i$ will be specified later. Although MAAIPM is still a second-order method, in our two block matrix-based accelerated algorithms, every iteration of solving the Newton direction has a time complexity of merely $O(m^2 \sum_{i=1}^{N} m_i + Nm^3)$ or $O(m \sum_{i=1}^{N} m_i^2 + \sum_{i=1}^{N} m_i^3)$, where a standard IPM would need $O\big((Nm + \sum_{i=1}^{N} m_i + 1)^2 (m \sum_{i=1}^{N} m_i + m)\big)$. For simplicity, let $m_i = m, \ i = 1, 2 \ldots, N$, then the time complexity of our algorithm in each iteration is $O(Nm^3)$, instead of standard IPM's complexity $O(N^3 m^4)$. Note that theoretically, when $Nm^2 = (Nm)^k$ for some $1 < k < 2$, the complexity of the standard IPM can be reduced to $O((Nm)^{\omega(k)}) + O((Nm)^3)$ via fast matrix computation methods, where the specific value of $\omega(k)$ can be found in table 3 of [20])

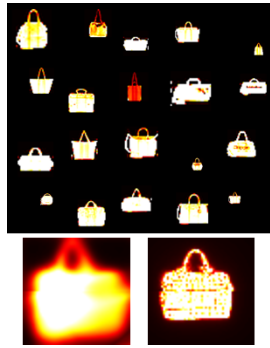

Figure 1: A comparison of algorithms for computing the barycenters between a Sinkhorn based approach[9](left) and MAAIPM(right). Samples of handbag(first 4 rows) are from Fashion-MNIST dataset.

**Practical effectiveness in speed and accuracy.** Compared to regularized methods, IPMs gain high-accuracy solutions and high convergence rate by nature. Numerical experiments show that our algorithm converges to highly accurate solutions of the original linear program with the least number of iterations. Figure 1 shows the advantages of our methods in accuracy in comparison to the well-developed Sinkhorn-type algorithm [14, 9].

There are more advantages of our approaches in real implementation. When the support points of measures are the same, there are several specially designed highly memory-efficient and thus very fast Sinkhorn based algorithms such as [46, 9]. However, when the support points of measures are different, the convolutional method in [46] is no longer applicable and the memory usage of our method is within a constant multiple of the popular memory-efficient first-order Sinkhorn method, IBP[9], much less than the memory used by a commercial solver. In this case, experiments also show that our algorithm can perform the best in both accuracy and overall runtime. Our algorithms also inherits a natural structure potentially fitting parallel computing scheme well. Those merits ensure that our algorithm is highly suitable for large-scale computation of Wasserstein barycenters.

The rest of the paper is organized as follows. In section 2, we briefly define the Wasserstein barycenter. In section 3, we present its linear programming formulation and introduce the IPM framework. In section 4, we present an IPM implementation that greatly reduces the computational cost of classical IPMs. In section 5, we present our numerical results.

## 2    Background and Preliminaries

In this section, we briefly recall the Wasserstein distance and the Wasserstein barycenter for a set of discrete probability measures [2, 16]. Let $\Sigma_n = \{\boldsymbol{a} \in \mathbb{R}^n | \sum_{i=1}^n a_i = 1, a_i \geq 0 \text{ for } i = 1, 2, \ldots n\}$ be the probability simplex in $\mathbb{R}^n$. For two vectors $\boldsymbol{s}^{(1)} \in \Sigma_{n_1}, \boldsymbol{s}^{(2)} \in \Sigma_{n_2}$, define the set of matrices $\mathcal{M}(\boldsymbol{s}^{(1)}, \boldsymbol{s}^{(2)}) = \{\Pi \in \mathbb{R}_+^{n_1 \times n_2} : \Pi \mathbf{1}_{n_2} = \boldsymbol{s}^{(1)}, \Pi^\top \mathbf{1}_{n_1} = \boldsymbol{s}^{(2)}\}$. Let $\mathcal{P} = \{(a_i, \boldsymbol{q}_i) : i = 1, \ldots, m\}$ denote the discrete probability measure supported on $m$ points $\boldsymbol{q}_1, \ldots, \boldsymbol{q}_m$ in $\mathbb{R}^d$ with weights $a_1, \ldots, a_m$ respectively. The Wasserstein barycenter of the two measures $\mathcal{U} = \{(a_i, \boldsymbol{q}_i) : i = 1, \ldots, m_1\}$ and $\mathcal{V} = \{(b_j, \boldsymbol{p}_j) : j = 1, \ldots, m_2\}$ is

$$\mathcal{W}_2(\mathcal{U}, \mathcal{V}) := \min \left\{ \sqrt{\sum_{i=1}^{m_1} \sum_{j=1}^{m_2} \pi_{ij} \|\boldsymbol{q}_i - \boldsymbol{p}_j\|^2} : \Pi = [\pi_{ij}] \in \mathcal{M}(\boldsymbol{a}, \boldsymbol{b}) \right\} \tag{1}$$

where $\boldsymbol{a} = (a_1, \ldots, a_{m_1})^\top$ and $\boldsymbol{b} = (b_1, \ldots, b_{m_2})^\top$. Consider a set of probability measures $\{\mathcal{P}^{(t)}, t = 1, \cdots, N\}$ where $\mathcal{P}^{(t)} = \{(a_i^{(t)}, \boldsymbol{q}_i^{(t)}) : i = 1, \ldots, m_t\}$, and let $\boldsymbol{a}^{(t)} = (a_1^{(t)}, \ldots, a_{m_t}^{(t)})^\top$. The Wasserstein barycenter (with $m$ support points) $\mathcal{P} = \{(w_i, \boldsymbol{x}_i) : i = 1, \cdots, m\}$ is another probability measure which is defined as a solution of the problem

$$\min_{\mathcal{P}} \frac{1}{N} \sum_{t=1}^N (\mathcal{W}_2(\mathcal{P}, \mathcal{P}^{(t)}))^2. \tag{2}$$

Furthermore, define the simplex $\mathcal{S} = \{(\boldsymbol{w}, \Pi^{(1)}, \ldots, \Pi^{(N)}) \in \mathbb{R}_+^m \times \mathbb{R}_+^{m \times m_1} \times \cdots \times \mathbb{R}_+^{m \times m_N} : \mathbf{1}_m^\top \boldsymbol{w} = 1, \boldsymbol{w} \geq 0; \Pi^{(t)} \mathbf{1}_{m_t} = \boldsymbol{w}, (\Pi^{(t)})^\top \mathbf{1}_m = \boldsymbol{a}^{(t)}, \Pi^{(t)} \geq 0, \forall t = 1, \cdots, N\}$. For a given set of support points $X = \{\boldsymbol{x}_1, \ldots, \boldsymbol{x}_m\}$, define the distance matrices $D^{(t)}(X) = [\|\boldsymbol{x}_i - \boldsymbol{q}_j^{(t)}\|_2^2] \in \mathbb{R}^{m \times m_t}$ for $t = 1, \ldots, N$. Then problem (2) is equivalent to

$$\min_{\boldsymbol{w}, X, \Pi^{(t)}} \sum_{t=1}^N \left\langle D^{(t)}(X), \Pi^{(t)} \right\rangle \text{ s.t. } (\boldsymbol{w}, \Pi^{(1)}, \ldots, \Pi^{(N)}) \in \mathcal{S}, \ \boldsymbol{x}_1, \ldots, \boldsymbol{x}_m \in \mathbb{R}^n. \tag{3}$$

Problem (3) is a nonconvex problem, where one needs to find the optimal support points X and the optimal weight vector $\boldsymbol{w}$ of a barycenter simultaneously. However, in many real applications, the support X of a barycenter can be specified empirically from the support points of $\{\mathcal{P}^{(t)}\}_{t=1}^N$. Indeed, in some cases, all measures in $\{\mathcal{P}^{(t)}\}_{t=1}^N$ have the same set of support points and hence the barycenter should also take the same set of support points. In view of this, we will also focus on the case when the support X is given. Consequently, problem (3) reduces to the following problem:

$$\min_{\boldsymbol{w}, \Pi^{(t)}} \sum_{t=1}^N \left\langle D^{(t)}, \Pi^{(t)} \right\rangle \text{ s.t. } (\boldsymbol{w}, \Pi^{(1)}, \ldots, \Pi^{(N)}) \in \mathcal{S} \tag{4}$$

where $D^{(t)}$ denotes $\mathcal{D}(X, Q^{(t)})$ for simplicity. In the following sections, we refer to problem (4) as the *Pre-specified Support Problem*, and call problem (3) the *Free Support Problem*.

# 3 General Framework for MAAIPM

**Linear programming formulation and preconditioning.** Note that the Pre-specified Support Problem is a linear program. In this subsection, we focus on removing redundant constraints. First, we vectorize the constraints $\Pi^{(t)}\mathbf{1}_{m_t} = \boldsymbol{w}$ and $\left(\Pi^{(t)}\right)^\top \mathbf{1}_m = \boldsymbol{a}^{(t)}$ captured in $\mathcal{S}$ to become

$$(\mathbf{1}_{m_t}^\top \otimes I_m)vec(\Pi^{(t)}) = \boldsymbol{w}, \ (I_{m_t} \otimes \mathbf{1}_m^\top)vec(\Pi^{(t)}) = \boldsymbol{a}^{(t)}, \ t = 1, \cdots, N.$$

Thus, problem (4) can be formulated into the standard-form linear program:

$$\min \ \boldsymbol{c}^\top \boldsymbol{x} \ \text{ s.t. } \ A\boldsymbol{x} = \boldsymbol{b}, \boldsymbol{x} \geq 0 \tag{5}$$

with $\boldsymbol{x} = (vec(\Pi^{(1)}); ...; vec(\Pi^{(N)}); \boldsymbol{w})$, $b = (\boldsymbol{a}^{(1)}; \boldsymbol{a}^{(2)}; ...; \boldsymbol{a}^{(N)}; \mathbf{0}_m; ...; \mathbf{0}_m; 1)$, $\boldsymbol{c} = (vec(D^{(1)}); ...; vec(D^{(N)}); \mathbf{0})$ and $A = \begin{bmatrix} E_1^\top & E_2^\top & 0 \\ 0 & E_3^\top & \mathbf{1}_m \end{bmatrix}^\top$, where $E_1$ is a block diagonal matrix: $E_1 = diag(I_{m_1} \otimes \mathbf{1}_m^\top, ..., I_{m_N} \otimes \mathbf{1}_m^\top)$; $E_2$ is a block diagonal matrix: $E_2 = diag(\mathbf{1}_{m_1}^\top \otimes I_m, ..., \mathbf{1}_{m_N}^\top \otimes I_m)$; and $E_3 = -\mathbf{1}_N \otimes I_m$. Let $M := \sum_{i=1}^N m_i$, $n_{row} := Nm + \sum_{i=1}^N m_i + 1$ and $n_{col} := m\sum_{i=1}^N m_i + m$. Then $A \in \mathbb{R}^{n_{row} \times n_{col}}, \boldsymbol{b} \in \mathbb{R}^{n_{row}}$ and $\boldsymbol{c} \in \mathbb{R}^{n_{col}}$. We are faced with a standard form linear program with $n_{col}$ variables and $n_{row}$ constraints. In the spacial case where all $m_i = m$, the number of variables is $O(Nm)$, and the number of constraints is $O(Nm^2)$.

For efficient implementations of IPMs for this linear program, we need to remove redundant constraints.

**Lemma 3.1** *Let $\bar{A} \in \mathbb{R}^{(n_{row}-N) \times n_{col}}$ be obtained from $A$ by removing the $(M+1)$-th, $(M+m+1)$-th, $\cdots$, $(M+(N-1)m+1)$-th rows of $A$, and $\bar{b} \in \mathbb{R}^{n_{row}-N}$ be obtained from $\boldsymbol{b}$ by removing the $(M+1)$-th, $(M+m+1)$-th, $\cdots$, $(M+(N-1)m+1)$-th entries of $\boldsymbol{b}$. Then 1) $\bar{A}$ has full row rank; 2) $\boldsymbol{x}$ satisfies $A\boldsymbol{x} = \boldsymbol{b}$ if and only if $\boldsymbol{x}$ satisfies $\bar{A}\boldsymbol{x} = \bar{b}$.*

The proof of this lemma is available in the supplement. With this lemma, the primal problem and dual problem of problem 5 can be written as

$$(\text{Primal}) \min \ \boldsymbol{c}^\top \boldsymbol{x} \ \text{ s.t. } \ \bar{A}\boldsymbol{x} = \bar{b}, \boldsymbol{x} \geq 0. \quad (\text{Dual}) \max \ \bar{b}^\top \boldsymbol{p} \ \text{ s.t. } \ \bar{A}^\top \boldsymbol{\lambda} + \boldsymbol{s} = \boldsymbol{c}, \boldsymbol{s} \geq 0. \tag{6}$$

**Framework of Matrix-based Adaptive Alternating Interior-point Method (MAAIPM).** When the support points are not pre-specified, we need to solve problem (3). As we just saw, When $X$ is fixed, the problem becomes a linear program. When $(\boldsymbol{w}, \{\Pi^{(t)}\})$ are fixed, the problem is a quadratic optimization problem with respect to $X$, and the optimal $X^*$ can be written in closed form as

$$\boldsymbol{x}_i^* = \left(\sum_{t=1}^N \sum_{j=1}^{m_t} \pi_{ij}^{(t)}\right)^{-1} \sum_{t=1}^N \sum_{j=1}^{m_t} \pi_{ij}^{(t)} \boldsymbol{q}_j^{(t)}, \quad i = 1, 2 \dots, m. \tag{7}$$

In anther word, (3) can be reformulated as

$$\min \ \boldsymbol{c}(\boldsymbol{x})^\top \boldsymbol{x} \ \text{ s.t.} \bar{A}\boldsymbol{x} = \bar{b}, \ \boldsymbol{x} \geq 0. \tag{8}$$

Since, as stated above, (3) is a non-convex problem and so it contains saddle points and local minima. This makes finding a global optimizer difficult. Examples of local minima and saddle points are available in the supplement. The alternating minimization strategy used in [16, 53, 54] alternates between optimizing $X$ by solving (7) and optimizing $(\boldsymbol{w}, \{\Pi^{(t)}\})$ by solving (4). However, this alternating approach cannot avoid local minima or saddle points. Every iteration may require solving a linear program (4), which is expensive when the problem size is large.

To overcome the drawbacks, we propose Matrix-based Adaptive Alternating IPM (MAAIPM). If the support is pre-specified, we solve a single linear program by predictor-corrector IPM[35, 40, 52]. If the support should be optimized, MAAIPM uses an adaptive strategy. At the beginning, because the primal variables are far from the optimal solution, MAAIPM updates $X^*$ of (7) after a few number of IPM iterations for $(\boldsymbol{w}, \{\Pi^{(t)}\})$. Then, MAAIPM updates $X^*$ after every IPM iteration and applies the "jump" tricks to escape local minima. Although MAAIPM cannot ensure finding a globally optimal solution, it can frequently get a better solution in shorter time. Since at the beginning MAAIPM updates $X^*$ after many IPM iterations, primal dual predictor-corrector IPM is more efficient. At the end, $X^*$ is updated more often and each update of $X^*$ changes the linear programming objective

function so that dual variables may be infeasible. However, the primal variables always remain feasible so that the primal IPM is more suitable at the end. Moreover, primal IPM is better for applying "jump" tricks or other local-minima-escaping techniques, which has been shown in [55]. Details and illustration are available in the supplement.

In predictor-corrector IPM, the main computational cost lies in solving the Newton equations, which can be reformulated as the normal equations

$$\bar{A}(D^k)^2 \bar{A}^\top \Delta \boldsymbol{\lambda}^k = \boldsymbol{f}^k, \tag{9}$$

where $D^k$ denotes $diag(x_i^{(k)}/s_i^{(k)})$ and $\boldsymbol{f}^k$ is in $\mathbb{R}^{n_{row}-N}$. This linear system of matrix $\bar{A}(D^k)^2\bar{A}^\top$ can be efficiently solved by the two methods proposed in the next section. In the primal IPM, MAAIPM combines following the central path with optimizing the support points, i.e., it contains three parts in one iteration, taking an Newton step in the logarithmic barrier function

$$\text{minimize} \quad \boldsymbol{c}^\top \boldsymbol{x} - \mu \sum_{i=1}^n \ln x_i, \quad \text{subject to} \quad \bar{A}\boldsymbol{x} = \boldsymbol{b}, \tag{10}$$

reducing the penalty $\mu$, and updating the support (7). The Newton direction $\boldsymbol{p}_k$ at the $k^{th}$ iteration is calculated by

$$\boldsymbol{p}^k = \boldsymbol{x}^k + (X^k)^2 \Big( \bar{A}^\top \big( \bar{A}(X^k)^2 \bar{A}^\top \big)^{-1} \big( \bar{A}(X^k)^2 \boldsymbol{c} - \mu \bar{A} X^k \mathbf{1} \big) - \boldsymbol{c} \Big) / \mu^k, \tag{11}$$

where $X^k = diag(x_i^{(k)})$. The main cost of primal IPM lies in solving a linear system of $\bar{A}(X^k)^2\bar{A}^\top$, which again can be efficiently solved by the two methods described in the following section. Further more, we also apply the warm-start technique to smartly choose the starting point of the next IPM after "jump" [45]. Compared with primal-dual IPMs' warm-start strategies [29, 28], our technique saves the searching time, and only requires slightly more memory. When we suitably set the termination criterion, numerical studies show that MAAIPM outperforms previous algorithms in both speed and accuracy, no matter whether the support is pre-specified or not.

## 4 Efficient Methods for Solving the Normal Equations

In this section, we discuss efficient methods for solving normal equations in the format $(\bar{A}D\bar{A}^\top)\boldsymbol{z} = \boldsymbol{f}$, where $D$ is a diagonal matrix with all diagonal entries being positive. Let $\boldsymbol{d} = diag(D)$, and $M_2 = N(m-1)$. First, through simple calculation, we have the following lemma on the structure of matrix $\bar{A}D\bar{A}^\top$, whose proof is available in the supplement.

**Lemma 4.1** $\bar{A}D\bar{A}^T$ *can be written in the following format:*

$$\bar{A}D\bar{A}^T = \begin{bmatrix} B_1 & B_2 & \mathbf{0} \\ B_2^\top & B_3 + B_4 & \boldsymbol{\alpha} \\ \mathbf{0} & \boldsymbol{\alpha}^\top & c \end{bmatrix}$$

*where $B_1 \in \mathbb{R}^{M \times M}$ is a diagonal matrix with positive diagonal entries; $B_2 \in \mathbb{R}^{M \times M_2}$ is a block-diagonal matrix with N blocks (the size of the i-th block is $(m-1) \times m_i$); $B_3 \in \mathbb{R}^{M_2 \times M_2}$ is a diagonal matrix with positive diagonal entries; Let $\boldsymbol{y} = \boldsymbol{d}(n_{col}-m+2 : n_{col})$, then $B_4 = (\mathbf{1}_N \mathbf{1}_N^\top) \otimes diag(\boldsymbol{y})$, and $\boldsymbol{\alpha} = -\mathbf{1}_N \otimes \boldsymbol{y}$; $c = \mathbf{1}_m^\top \boldsymbol{d}(n_{col} - m + 1 : n_{col})$.*

**Single low-rank regularization method (SLRM).** Briefly speaking, we will perform several basic transformations on the matrix $\bar{A}D\bar{A}^T$ to transform it into an easy-to-solve format. Then we solve the system with the transformed coefficient matrix and finally transform the obtained solution back to get an solution of $(\bar{A}D\bar{A}^\top)\boldsymbol{z} = \boldsymbol{f}$.

Define $V_1 := \begin{bmatrix} I_M & & \\ -B_2^\top B_1^{-1} & I_{M_2} & \\ & & 1 \end{bmatrix}$, $V_2 := \begin{bmatrix} I_M & & \\ & I_{M_2} & -\boldsymbol{\alpha}/c \\ & & 1 \end{bmatrix}$, $A_1 := B_3 - B_2^\top B_1^{-1} B_2$ and

$A_2 := B_4 - \frac{1}{c}\boldsymbol{\alpha}\boldsymbol{\alpha}^\top$. Then,

$$V_2 V_1 \bar{A}D\bar{A}^T V_1^\top V_2^\top = \begin{bmatrix} B_1 & & \\ & B_3 - B_2^\top B_1^{-1} B_2 + B_4 - \frac{1}{c}\boldsymbol{\alpha}\boldsymbol{\alpha}^\top & \\ & & c \end{bmatrix} = \begin{bmatrix} B_1 & & \\ & A_1 + A_2 & \\ & & c \end{bmatrix}.$$

Define $Y = diag(\boldsymbol{y}) - \frac{1}{c}\boldsymbol{y}\boldsymbol{y}^\top$, we have the following lemma.

**Lemma 4.2**

*a)* $A_1$ *is a block-diagonal matrix with N blocks. The size of each block is* $(m-1) \times (m-1)$. *Further more,* $A_1$ *is positive definite and strictly diagonal dominant. b)* $A_2 = (\mathbf{1}_N \mathbf{1}_N^\top) \otimes Y$, *and Y is positive definite and strictly diagonal dominant.*

Since the positive definiteness and diagonal dominance claimed in this lemma, the computation of the inverse matrices of each block of $A_1$ and $A_2$ is numerically stable. Now we introduce the procedure for solving $(\bar{A}D\bar{A}^T)\boldsymbol{z} = \boldsymbol{f}$, as descried in Algorithm 1 ($\boldsymbol{z}^{(1)}$ - $\boldsymbol{z}^{(4)}$ in the algorithm are intermediate variables). In step 7, we need to solve a linear system with coeffi-

---

**Algorithm 1:** Solver for the normal equation $(\bar{A}D\bar{A}^T)\boldsymbol{z} = \boldsymbol{f}$

---

**Input:** $\boldsymbol{d} = diag(D) \in \mathbb{R}^{n_{col}}$; $\boldsymbol{f} \in \mathbb{R}^{M+N(m-1)+1}$

1 compute $B_1, B_2, B_3$, vector $\boldsymbol{y} = \boldsymbol{d}(n_{col} - m + 2 : n_{col})$ and $c$;
2 compute $T = B_2^\top B_1^{-1}$ and matrices $V_1, V_2$;
3 compute $A_1 = B_3 - TB_2$ and $A_2 = (\mathbf{1}_N \mathbf{1}_N^\top) \otimes (diag(\boldsymbol{y}) - \frac{1}{c}\boldsymbol{y}\boldsymbol{y}^\top)$;
4 compute $\boldsymbol{z}^{(1)} = V_1 \boldsymbol{f}$ and $\boldsymbol{z}^{(2)} = V_2 \boldsymbol{z}^{(1)}$;
5 compute $\boldsymbol{z}^{(3)}(1:M) = B_1^{-1}\boldsymbol{z}^{(2)}(1:M)$;
6 compute $\boldsymbol{z}^{(3)}(M + M_2 + 1) = \frac{1}{c}\boldsymbol{z}^{(2)}(M + M_2 + 1)$;
7 solve the linear system with coefficient matrix $A_1 + A_2$ to get
$\boldsymbol{z}^{(3)}(M+1:M+M_2) = (A_1 + A_2)^{-1}\boldsymbol{z}^{(2)}(M+1:M+M_2)$;
8 compute $\boldsymbol{z}^{(4)} = V_2^\top \boldsymbol{z}^{(3)}$, $\boldsymbol{z} = V_1^\top \boldsymbol{z}^{(4)}$;

**Output:** $\boldsymbol{z}$

---

cient matrix of dimension $N(m-1) \times N(m-1)$, which is hard to compute with common methods for dense symmetric matrices. In view of the low-rank structure of the matrix $A_2$, we introduce a method, namely Single Low-rank Regularization Method (SLRM), which requires only $O(Nm^3)$ flops in computation. Assume $A_1 = diag(A_{11}, A_{22}, ..., A_{NN})$ and define $U = \begin{bmatrix} I_{N-1} & \mathbf{1}_{N-1} \\ 0 & 1 \end{bmatrix} \otimes I_{m-1}$.

We can solve the linear system $(A_1 + A_2)\boldsymbol{x} = \boldsymbol{g}$ by Algorithm 2.

The proof of correctness of Algorithm 2 and other analysis is available in the supplement.

**Double low-rank regularization method (DLRM) when** $m$ **is large.** In many applications, $m$ is relatively large compared to $m_t$. For instance, in the area of image identification, the pixel support points of the images at hand are sparse (small $m_t$) but different. To find the "barycenter" of these images, we need to assume the "barycenter" image has much more pixel support points (large $m$) than all the sample images. Sometimes, $m$ might be about 5 to 20 times of each $m_t$. In this case, the computational cost of step 1 in SLRM is

---

**Algorithm 2:** SLRM for the system $(A_1 + A_2)\boldsymbol{x} = \boldsymbol{g}$

---

**Input:** $A_1$, $A_2$, $\boldsymbol{g}$

1 compute $A_{ii}^{-1}, i = 1, .., N$;
2 set $A_1^{-1} = diag(A_{11}^{-1}, ..., A_{NN}^{-1})$;
3 compute $\boldsymbol{x}^{(1)} = A_1^{-1}\boldsymbol{g}$;
4 compute $\boldsymbol{x}^{(2)} = U^T \boldsymbol{x}^{(1)}$;
5 compute $\boldsymbol{x}^{(3)}(end - m + 2 : end) = (Y^{-1} + \sum_{i=1}^N A_{ii}^{-1}) \backslash \boldsymbol{x}^{(2)}(end - m + 2 : end)$;
6 set $\boldsymbol{x}^{(3)}(1 : end - m + 1) = 0$;
7 compute $\boldsymbol{x}^{(4)} = U\boldsymbol{x}^{(3)}$ and $\boldsymbol{x}^{(5)} = A_1^{-1}\boldsymbol{x}^{(4)}$;
8 compute $\boldsymbol{x} = \boldsymbol{x}^{(1)} - \boldsymbol{x}^{(5)}$;

**Output:** $\boldsymbol{x}$

---

heavy, since we need to solve $N$ linear systems with dimension $m \times m$. In this subsection, we use the low rank regularization formula to further reduce the computational cost.

In view of lemma 4.1, assume

$$B_1 = diag(B_{11}, ..., B_{1N}), \quad B_2 = diag(B_{21}, ..., B_{2N}), \quad B_3 = diag(B_{31}, ..., B_{3N}).$$

where $B_{1i} \in \mathbb{R}^{m_i \times m_i}$, $B_{2i} \in \mathbb{R}^{m_i \times (m-1)}$ and $B_{3i} \in \mathbb{R}^{(m-1) \times (m-1)}$. Recall that $A_1 = B_3 - B_2^\top B_1^{-1}B_2$ and $A_1 = diag(A_{11}, ..., A_{NN})$, we have $A_{ii} = B_{3i} - B_{2i}^\top B_{1i}^{-1}B_{2i}$. Since $m >> m_i$, we can use the following formula:

$$A_{ii}^{-1} = (B_{3i} - B_{2i}^\top B_{1i}^{-1}B_{2i})^{-1} = B_{3i}^{-1} + B_{3i}^{-1}B_{2i}^\top (B_{1i} - B_{2i}B_{3i}^{-1}B_{2i}^\top)^{-1}B_{2i}B_{3i}^{-1}. \quad (12)$$

Instead of calculating and storing each $A_{ii}$ explicitly, we can just calculate and store each $(B_{1i} - B_{2i}B_{3i}^{-1}B_{2i}^\top)^{-1}$. When we need to calculate $A_{ii}\boldsymbol{y}$ for some vector $\boldsymbol{y}$, we can use (12) and sequentially

multiply each matrix with vectors. As a result, the flops required in step 1 of SLRM reduce to $O(m\Sigma_{i=1}^{N} m_i^2 + \Sigma_{i=1}^{N} m_i^3)$, and the total memory usage of whole MAAIPM is $O(m\Sigma_{i=1}^{N} m_i)$, which is at the same level (except for a constant) of a primal variable.

**Complexity analysis.** The following theorem summarizes the time and space complexity of the aforementioned two methods.

**Theorem 4.3** *a) For SLRM, the time complexity in terms of flops is $O(m^2 \sum_{i=1}^{N} m_i + Nm^3)$, and the memory usage in terms of doubles is $O(m \sum_{i=1}^{N} m_i + Nm^2)$; b) For the DLRM, the time complexity in terms of flops is $O(m \sum_{i=1}^{N} m_i^2 + \sum_{i=1}^{N} m_i^3)$, and the memory usage in terms of doubles is $O(m \sum_{i=1}^{N} m_i + \sum_{i=1}^{N} m_i^2)$.*

We can choose between SLRM and DLRM for different cases to achieve lower time and space complexity. Note that as $N, m, m_i$ grows up, the memory usage here is within an constant time of the representative Sinkhorn type algorithms like IBP[9].

## 5 Experiments

We conduct three numerical experiments to investigate the real performance of our methods. The first experiment shows the advantages of SLRM and DLRM over traditional approaches in solving Newton equations with a same structure as barycenter problems. The second experiment fully demonstrates the merits of MAAIPM: high speed/accuracy and more efficient memory usage. In the last experiment with real benchmark data, MAAIPM recovers the images better than any other approach implemented. In different experiments, we compare our methods with state-of-art commercial solvers(MATLAB, Gurobi, MOSEK), the iterative Bregman projection (IBP) by [9], Bregman ADMM (BADMM) [51, 54]. The result also illustrates MAAIPM's superiority over symmetric Gauss-Seidel ADMM (sGS-ADMM) [53].

All experiments are run in Matlab R2018b on a workstation with two processors, Intel(R) Xeon(R) Processor E5-2630@2.40Ghz (8 cores and 16 threads per processor) and 64GB of RAM, equipped with 64-bit Windows 10 OS. Full experiment details are available in the supplement.

**Experiments on solving the normal equations:** For figure 2, one can see that both SLRM and DLRM clealy outperform the Matlab solver in in all cases. For computation time, SLRM increases linearly with respect to $N$ and $m'$, and DLRM increases linearly with respect to $N$ and $m$, which matches the conclusions in Theorem 4.3. In practice, we select SLRM when $m^2 \leq 4 \sum_{t=1}^{N} m_t^2$ and DLRM when $m^2 > 4 \sum_{t=1}^{N} m_t^2$.

**Experiments on barycenter problems:** In this experiment, we set $d = 3$ for convenience. For $\mathcal{P}^{(t)}$, each entry of $(\boldsymbol{q}_1^{(t)}, \ldots, \boldsymbol{q}_{m'}^{(t)})$ is generated with i.i.d. standard Gaussian distribution. The entries of the weight vectors $(a_1^{(t)}, \ldots, a_{m'}^{(t)})$ are simulated by uniform distribution on $(0, 1)$ and then are normalized. Next we apply the $k$-means[2] method to choose $m$ points to be the support points. Note that Gurobi and MOSEK use a crossover strategy when close to the exact solution to ensure obtaining a highly accurate solution, we can regard Gurobi's objective value $\mathcal{F}_{gu}$ as the exact optimal value of the linear program (4).

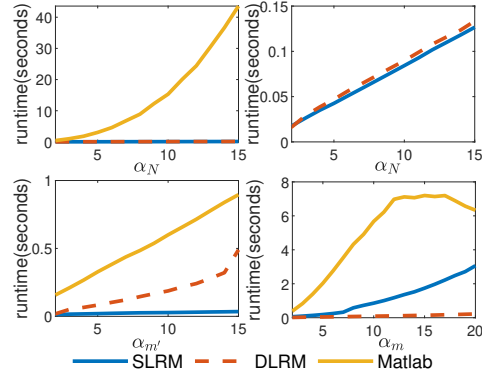

Figure 2: Average computation time of 200 independent trials in solving the linear system. Entries of diagonal $D$ and $\boldsymbol{f}$ are generated by uniform distribution in $(0, 1)$. In base situation, $N = 50, m = 50, m' = 25$. Sub-figures show the computation times when rescaling $N$, $m$ and $m_1 = \cdots = m_N = m'$ by respectively $\alpha_N$, $\alpha_m$ and $\alpha_{m'}$ times.

Let "normalized obj" denote the normalized objective value defined by $|\mathcal{F}_{method} - \mathcal{F}_{gu}|/\mathcal{F}_{gu}$, where $\mathcal{F}_{method}$ is the objective value respectively obtained by each method. Let "feasibility error" denote

$\max\left\{\frac{\|\{\Pi^{(t)}\mathbf{1}_{m_t}-\boldsymbol{w}\}\|_F}{1+\|\boldsymbol{w}\|_F+\|\{\Pi^{(t)}\}\|_F}, \frac{\|\{(\Pi^{(t)})^\top\mathbf{1}_m-\boldsymbol{a}^{(t)}\}\|_F}{1+\|\{\boldsymbol{a}^{(t)}\}\|_F+\|\{\Pi^{(t)}\}\|_F}, |\mathbf{1}^\top\boldsymbol{w}-1|\right\}$, as a measure of the distance to the feasible set.

From figure 3, we see that MAAIPM displays a super-linear convergence rate for the objective, which is consistent with the result of [56]. Note that the feasibility error of MAAIPM increases a little bit near the end but is still much lower than BADMM and IBP. Although other methods may have lower objective values in early stages, their solutions are not acceptable due to high feasibility errors.

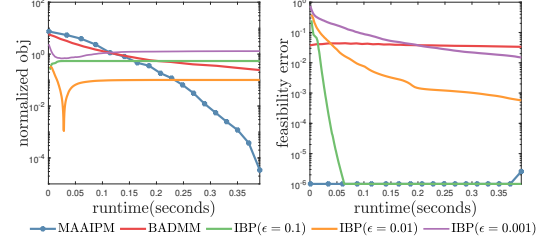

Figure 3: Performance of methods in pre-specified support cases. $N = m = 50$ and $m_1 = \cdots = m_N = 50$

Then we run numerical experiments to test the computation time of methods in pre-specified support points cases. For MAAIPM, we terminate it when $(\boldsymbol{b}^\top\boldsymbol{\lambda}_k - \boldsymbol{c}^\top\boldsymbol{x}_k)/(1 + |\boldsymbol{b}^\top\boldsymbol{\lambda}_k| + |\boldsymbol{c}^\top\boldsymbol{x}_k|)$ is less than $5\times 10^{-5}$. For sGS-ADMM, we compare with it indirectly by the benchmark claimed in their paper [53]: commercial solver Gurobi 8.1.0 [24] (academic license) with the default parameter settings. We also compare with another commercial solver MOSEK 9.1.0(academic license). In our observation, MAAIPM can frequently perform better than other popular commercial solvers. We use the default parameter setting(optimal for most cases) for Gurobi and MOSEK so that they can exploit multiple processors (16 threads) while other methods are implemented with only one thread[3]. For BADMM, we follow the algorithm 4 in [54] to implement and terminate when $\|\Pi^{(k,1)} - \Pi^{(k,2)}\|_F/(1 + \|\Pi^{(k,1)}\|_F + \|\Pi^{(k,2)}\|_F) < 10^{-5}$. Set $\|\{A_t\}\|_F = \left(\sum_{t=1}^N \|A_t\|_F^2\right)^{\frac{1}{2}}$. For IBP, we follow the remark 3 in [9] to implement the method, terminate it when $\|\{u_k^{(n)}\} - \{u_k^{(n-1)}\}\|_F/(1 + \|\{u_k^{(n)}\}\|_F + \|\{u_k^{(n-1)}\}\|_F) < 10^{-8}$ and $\|\{v_k^{(n)}\} - \{v_k^{(n-1)}\}\|_F/(1 + \|\{v_k^{(n)}\}\|_F + \|\{v_k^{(n-1)}\}\|_F) < 10^{-8}$, and choose the regularization parameter $\epsilon$ from $\{0.1, 0.01, 0.001\}$ in our experiments. For BADMM and IBP, we implement the Matlab codes[4] by J.Ye et al. [54] and set the maximum iterate number respectively 4000 and $10^5$.

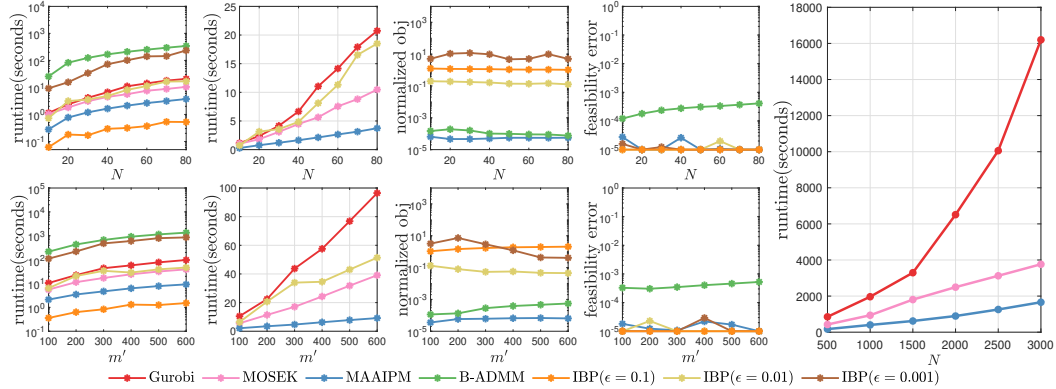

Figure 4: The left 8 figures are the average computation time, normalized objective value and feasibility error of Gurobi, MOSEK, MAAIPM, BADMM and IBP($\epsilon = 0.1, 0.01, 0.001$) in pre-specified support cases from 30 independent trials. In the first row, $m = 100$, $m_t$ follows an uniform distribution on $(75, 125)$. In the second row, $N = 50$, $m = 100$ and $m_1 = \cdots = m_N = m'$. The right figure is the average computation time of Gurobi and MAAIPM in pre-specified support cases from 10 independent trials. $m_t$ follows a uniform distribution on $(150, 250)$, and $m = 200$.

From the left 8 sub-figures in figure 4 one can observe that MAAIPM returns a considerably accurate solution in the second shortest computation time. For IBP, although it returns an objective value in the shortest time when $\epsilon = 0.1$, the quality of the solution is almost the worst. Because IBP only solves an approximate problem, if $\epsilon$ is set smaller, the computation time sharply increases but the

quality of the solution is still not ensured. For BADMM, it gives a solution close to the exact one, but requires much more computation time.

For Gurobi and MOSEK, although they can exploit 16 threads, the computation time is far more than that of MAAIPM That is to say, MAAIPM also largely outperforms sGS-ADMM in speed, according to table 1, 2, 3 in [53]. Moreover, because the number of iterations remains almost independent of the problem size, the main computational cost of MAAIPM is approximately linear with respect to $N$ and $m'$. In fact, when $N = 5000$, MAAIPM requires only 3098.23 seconds, while MOSEK uses over 20000 seconds. Although the memory usage of MAAIPM is within a constant multiple of that of IBP, the former one is ususaly larger than the latter one. But the right subfigure in Figure 4 and the case of $N = 5000$ demonstrate that MAAIPM's memory usage

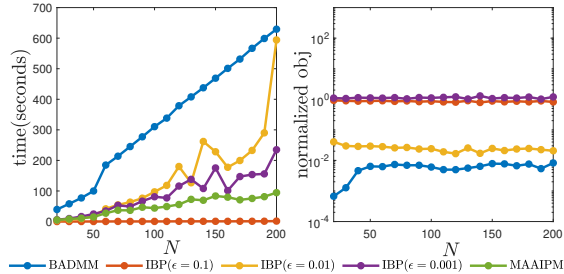

Figure 5: computation time and normalized objective value of MAAIPM, BADMM and IBP in the free support cases from 30 independent trials. "Normalized obj" denote $\mathcal{F}_{method}/\mathcal{F}_{MAAIPM} - 1$, where $\mathcal{F}_{method}$ is the objective value obtained by each method. $N$ takes different values and $m = m' = 50$.

is managed more efficient compared to Gurobi and MOSEK. These positive traits are consistent with the time and memory complexity proved in Theorem 4.3.

Next, we conduct numerical studies to test MAAIPM in free support cases, i.e., problem (3). Same as [54], we implement the version of BADMM and IBP that can automatically update support points and set the initial support points in multivariate normal distribution. We set the maximum number of iterations in BADMM and IBP as $10^4$ and $10^6$. The entries of $(\boldsymbol{q}_1^{(t)}, \dots, \boldsymbol{q}_{m'}^{(t)})$ are generated with i.i.d. uniform distribution in $(0, 100)$ and the initial support points follows a Gaussian distribution. In figure 6, "Normalized obj" denotes $\mathcal{F}_{method}/\mathcal{F}_{MAAIPM} - 1$, where $\mathcal{F}_{method}$ is the objective value obtained by each iteration of methods. From figure 5 and 6, one can see that, in the free support cases, MAAIPM can still obtain the smallest objective value in the second shortest time. That is because MAAIPM updates support more frequently and adopts "jump" tricks to avoid the local minima. Although IBP can obtain an approximate value in the shortest time when $\epsilon = 0.1$, the quality of the barycenter is too low to be useful.

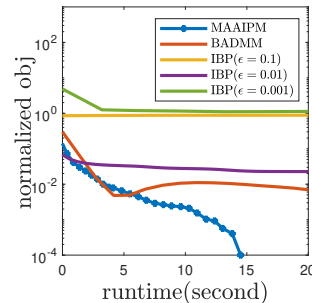

Figure 6: Performance of methods in free support cases. $N = 40$, $m = m_1 = m_2 = \cdots = m_N = 50$.

**Experiments on real applications:** We conduct similar experiments to [16, 53] on the MNIST[4] and Fashion-MNIST[4] datasets. In MNIST, We randomly select 200 images for digit 8 and resize each image to 0.5, 1, 2 times of its original size $28 \times 28$. In Fashion-MNIST, we randomly select 20 images of handbag, and resize each image to 0.5, 1 time of the original size. The support points of images are dense and different. Next, for

Table 1: Experiments on datasets

| | MNIST | | | Fashion-MNIST | | |
|---|---|---|---|---|---|---|
| time(seconds) | 250 | 500 | 1000 | 25 | 50 | 75 |
| MAAIPM | | | | | | |
| BADMM | | | | | | |
| IBP($\epsilon = 0.01$) | | | | | | |

each case, we apply MAAIPM, BADMM and IBP($\epsilon = 0.01$) to compute the Wasserstein barycenter in respectively free support cases and pre-specified support cases. From table 1, one can see that, MAAIPM obtained the clearest and sharpest barycenters within the least computation time.

## Acknowledgments

We thank Tianyi Lin, Simai He, Bo Jiang, Qi Deng and Yibo Zeng for helpful discussions and fruitful suggestions.

## Footnotes

*Haoyue Wang and Zikai Xiong are corresponding authors.

[2]We call the Malab function "kmeans" in statistics and machine learning toolbox.

[3]We call the Matlab function "maxNumCompThreads(1)"

[4]Available in https://github.com/bobye/WBC_Matlab

[4] Available in http://yann.lecun.com/exdb/mnist/ and https://github.com/zalandoresearch/fashion-mnist

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
