[Supplementary Material · MAAIPMbarycenter_supplement.pdf]

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

# A   Proof of lemma 3.1

To justify the claims in lemma 3.1, we follows the following line of proof: a) First, we show that through a series of row transformations, we can transform matrix A into a matrix whose elements in $(M+1)$-th, $(M+m+1)$-th, $\cdots$, $(M+(N-1)m+1)$-th rows are zeros, and elements in other positions are the same as A. b) Second, we prove that the matrix $\bar{A}$ has full row rank.

a). From the definition of matrix A, we have

$$
A = \begin{bmatrix}
F_1 & & & & \\
 & F_2 & & & \\
 & & \ddots & & \\
 & & & F_N & \\
G_1 & & & & -I_m \\
 & G_2 & & & -I_m \\
 & & \ddots & & \vdots \\
 & & & G_N & -I_m \\
 & & & & \mathbf{1}_m^T
\end{bmatrix}
\tag{13}
$$

where $F_i = I_{m_i} \otimes \mathbf{1}_m^\top$, $G_i = \mathbf{1}_{m_i}^\top \otimes I_m$ for $i = 1, ..., N$.

Let

$$
\boldsymbol{e}_1 = \begin{bmatrix} 1 \\ 0 \\ \vdots \\ 0 \end{bmatrix}_{m \times 1}, \quad
T_i = \begin{bmatrix} 1 & 1 & \cdots & 1 \\ & & & \\ & & & \end{bmatrix}_{m \times m_i}, \quad
S_i = \begin{bmatrix} 1 & 1 & \cdots & 1 \\ & 1 & & \\ & & \ddots & \\ & & & 1 \end{bmatrix}_{m \times m}, \quad i = 1, ..., N
$$

and

$$
L_1 = \begin{bmatrix} I_{m_1} & & & & & \\ & \ddots & & & & \\ & & I_{m_N} & & & \\ -T_1 & & & I_m & & \\ & \ddots & & & \ddots & \\ & & -T_N & & & I_m \\ & & & & & & 1 \end{bmatrix}, \quad L_2 = \begin{bmatrix} I_{m_1} & & & & & \\ & \ddots & & & & \\ & & I_{m_N} & & & \\ & & & S_1 & & & e_1 \\ & & & & \ddots & & \vdots \\ & & & & & S_N & e_1 \\ & & & & & & 1 \end{bmatrix}
$$

Then

$$
L_2 L_1 A = \begin{bmatrix} F_1 & & & & & & \\ & F_2 & & & & & \\ & & \ddots & & & & \\ & & & F_N & & & \\ G_1^{(1)} & & & & & & H^{(1)} \\ & G_2^{(1)} & & & & & H^{(1)} \\ & & \ddots & & & & \vdots \\ & & & G_N^{(1)} & & & H^{(1)} \\ & & & & & & \mathbf{1}_m^T \end{bmatrix},
$$

where $G_i^{(1)} = S_i G_i - S_i T_i F_i$, $H^{(1)} = e_1 \mathbf{1}_m^\top - S_i$. It is easy to verify that elements in the first rows of $H^{(1)}$ and $G_i^{(1)}$, $i = 1, ..., N$ are zeros. We have proved the claims in a).

b). As defined in the claims of lemma 3.1, $\bar{A}$ is obtained by removing the $(M+1)$-th, $(M+m+1)$-th, $\cdots$, $(M+(N-1)m+1)$-th rows of A. That is,

$$
\bar{A} = \begin{bmatrix} F_1 & & & & & & \\ & F_2 & & & & & \\ & & \ddots & & & & \\ & & & F_N & & & \\ G_1^{(2)} & & & & & & H^{(2)} \\ & G_2^{(2)} & & & & & H^{(2)} \\ & & \ddots & & & & \vdots \\ & & & G_N^{(2)} & & & H^{(2)} \\ & & & & & & \mathbf{1}_m^\top \end{bmatrix}
$$

where $G_i^{(2)} = G_i^{(1)}(2:m,:) = \mathbf{1}_{m_i}^\top \otimes [\mathbf{0}_{m-1}, I_{m-1}]$, $H^{(2)} = H^{(1)}(2:m,:) = [\mathbf{0}_{m-1}, -I_{m-1}]$ and $F_i = I_{m_i} \otimes \mathbf{1}_m^\top$. Let $n'_{row} = M + N(m-1) + 1$.

For $i = 1, ..., N$, let

$$
U_i = I_{m_i} \otimes \begin{bmatrix} 1 & -1 & \cdots & -1 \\ & 1 & & \\ & & \ddots & \\ & & & 1 \end{bmatrix}_{m \times m}, \quad U_{N+1} = \begin{bmatrix} 1 & -1 & \cdots & -1 \\ & 1 & & \\ & & \ddots & \\ & & & 1 \end{bmatrix}_{m \times m}, \quad R_1 = \begin{bmatrix} U_1 & & & \\ & U_2 & & \\ & & \ddots & \\ & & & U_{N+1} \end{bmatrix}
$$

then

$$
\bar{A}R_1 =
\begin{bmatrix}
F_1^{(3)} & & & & & \\
& F_2^{(3)} & & & & \\
& & \ddots & & & \\
& & & F_N^{(3)} & & \\
G_1^{(3)} & & & & H^{(3)} \\
& G_2^{(3)} & & & H^{(3)} \\
& & \ddots & & \vdots \\
& & & G_N^{(3)} & H^{(3)} \\
& & & & \boldsymbol{\alpha}^\top
\end{bmatrix}
$$

where $F_i^{(3)} = F_i U_i = I_{m_i}\otimes[1,\mathbf{0}_{m-1}^\top], G_i^{(3)} = G_i^{(2)}U_i = G_i^{(2)} = \mathbf{1}_{m_i}^\top\otimes[\mathbf{0}_{m-1}, I_{m-1}], i = 1,...,N,$
$H^{(3)} = H^{(2)}U_{N+1} = H^{(2)} = [\mathbf{0}_{m-1}, -I_{m-1}]$ and $\boldsymbol{\alpha}^\top = \mathbf{1}_m^\top U_{N+1} = [1, \mathbf{0}_{m-1}^\top].$

Let

$$
\tilde{K} =
\begin{bmatrix}
0 & & & \\
& -1 & & \\
& & \ddots & \\
& & & -1
\end{bmatrix}_{m\times m}
, \quad
K_i =
\begin{bmatrix}
I_m & \tilde{K} & \cdots & \tilde{K} \\
& I_m & & \\
& & \ddots & \\
& & & I_m
\end{bmatrix}_{mm_i\times mm_i}
, \quad
R_2 =
\begin{bmatrix}
K_1 & & & \\
& \ddots & & \\
& & K_N & \\
& & & I_m
\end{bmatrix}
$$

then

$$
\bar{A}R_1R_2 =
\begin{bmatrix}
F_1^{(4)} & & & & & \\
& F_2^{(4)} & & & & \\
& & \ddots & & & \\
& & & F_N^{(4)} & & \\
G_1^{(4)} & & & & H^{(3)} \\
& G_2^{(4)} & & & H^{(3)} \\
& & \ddots & & \vdots \\
& & & G_N^{(4)} & H^{(3)} \\
& & & & \boldsymbol{\alpha}^\top
\end{bmatrix}
$$

where $F_i^{(4)} = F_i^{(3)}K_i = F_i^{(3)} = I_{m_i} \otimes [1,\mathbf{0}_{m-1}^\top], G_i^{(4)} = G_i^{(3)}K_i = [\mathbf{0}_{m-1}, I_{m-1}, \mathbf{0}_{(m-1)\times(mm_i-m)}], i = 1,...,N,$

Let $\tilde{A}$ be the matrix composing of the first $(mM + 1)$ columns of $\bar{A}R_1R_2$. That is,

$$
\tilde{A} =
\begin{bmatrix}
F_1^{(4)} & & & & \\
& F_2^{(4)} & & & \\
& & \ddots & & \\
& & & F_N^{(4)} & \\
G_1^{(4)} & & & & \\
& G_2^{(4)} & & & \\
& & \ddots & & \\
& & & G_N^{(4)} & \\
& & & & 1
\end{bmatrix}
$$

Matrix $\tilde{A}$ satisfies two properties:

(1) Each row of $\tilde{A}$ has one and only one nonzero element (being 1) with other elements being 0;

(2) Each column of $\tilde{A}$ has at most one nonzero element.

Therefore, there exists permutation matrices $P_1 \in \mathbb{R}^{n'_{row}}$ and $Q_1 \in \mathbb{R}^{n_{col}-m+1}$ such that $P_1 \tilde{A} Q_1 = [I_{n'_{row}}, 0_{n'_{row} \times (Mm+1)}]$. Thus $rank(\tilde{A}) = rank(P_1 \tilde{A} Q_1) = n'_{row}$ and $rank(\bar{A}) = n'_{row}$.

## B   Proof of lemma 4.1

In this subsection, we give the proof of lemma 4.1.

*Proof.* Let $\boldsymbol{d}$ be the diagonal vector of matrix $D$; $M := \sum_{i=1}^{N} m_i$ and $M_2 := N(m-1)$. Same as the preceding section, the structure of $\bar{A}$ as:

$$
\bar{A} = \begin{bmatrix}
F_1 & & & & & \\
& F_2 & & & & \\
& & \ddots & & & \\
& & & F_N & & \\
G_1^{(2)} & & & & & H^{(2)} \\
& G_2^{(2)} & & & & H^{(2)} \\
& & \ddots & & & \vdots \\
& & & G_N^{(2)} & & H^{(2)} \\
& & & & & \mathbf{1}_m^\top
\end{bmatrix}
$$

where $G_i^{(2)} = G_i^{(1)}(2:m,:) = \mathbf{1}_{m_i}^\top \otimes [\mathbf{0}_{m-1}, I_{m-1}]$, $H^{(2)} = H^{(1)}(2:m,:) = [\mathbf{0}_{m-1}, -I_{m-1}]$ and $F_i = I_{m_i} \otimes \mathbf{1}_m^\top$.

Let

$$
\bar{A}_1 := \bar{A}(1:M,:) = \begin{bmatrix}
F_1 & & & \\
& F_2 & & \\
& & \ddots & \\
& & & F_N
\end{bmatrix},
$$

$$
\bar{A}_2 := \bar{A}(M+1:M+(m-1)N,:) = \begin{bmatrix}
G_1^{(2)} & & & & H^{(2)} \\
& G_2^{(2)} & & & H^{(2)} \\
& & \ddots & & \vdots \\
& & & G_N^{(2)} & H^{(2)}
\end{bmatrix},
$$

$$
\bar{A}_3 := \bar{A}(M+(m-1)N+1,:) = \begin{bmatrix} & & \mathbf{1}_m^\top \end{bmatrix}.
$$

Then

$$
\bar{A} = \begin{bmatrix} \bar{A}_1 \\ \bar{A}_2 \\ \bar{A}_3 \end{bmatrix} \quad and \quad \bar{A} D \bar{A}^\top = \begin{bmatrix}
\bar{A}_1 D \bar{A}_1^\top & \bar{A}_1 D \bar{A}_2^\top & \bar{A}_1 D \bar{A}_3^\top \\
\bar{A}_2 D \bar{A}_1^\top & \bar{A}_2 D \bar{A}_2^\top & \bar{A}_2 D \bar{A}_3^\top \\
\bar{A}_3 D \bar{A}_1^\top & \bar{A}_3 D \bar{A}_2^\top & \bar{A}_3 D \bar{A}_3^\top
\end{bmatrix}.
$$

Now we analyze the structure of each sub-matrix $\bar{A}_i D \bar{A}_j^\top$ and rename them for conciseness. Let

$$
D = \begin{bmatrix}
D_1 & & & \\
& D_2 & & \\
& & \ddots & \\
& & & D_{N+1}
\end{bmatrix},
$$

where $D_i \in \mathbb{R}^{mm_i \times mm_i}$, $i = 1, \ldots, N$ and $D_{N+1} \in \mathbb{R}^{m \times m}$. Then

$$
\bar{A}_1 D \bar{A}_1^\top = \begin{bmatrix}
F_1 D_1 F_1^\top & & \\
& \ddots & \\
& & F_N D_N F_N^\top
\end{bmatrix} := B_1.
$$

Each $F_i D_i F_i^\top$ is a diagonal matrix with positive diagonal entries.

$$\bar{A}_2 D \bar{A}_1^\top = \begin{bmatrix} G_1^{(2)} D_1 F_1^\top & & \\ & \ddots & \\ & & G_N^{(2)} D_N F_N^\top \end{bmatrix} := B_2^\top,$$

$$\bar{A}_2 D \bar{A}_2^\top = \begin{bmatrix} G_1^{(2)} D_1 G_1^{(2)\top} & & \\ & \ddots & \\ & & G_N^{(2)} D_N G_N^{(2)\top} \end{bmatrix} + \begin{bmatrix} H^{(2)} D_{N+1} H^{(2)\top} & \cdots & H^{(2)} D_{N+1} H^{(2)\top} \\ \vdots & & \vdots \\ H^{(2)} D_{N+1} H^{(2)\top} & \cdots & H^{(2)} D_{N+1} H^{(2)\top} \end{bmatrix}.$$

(14)

where $H^{(2)} D_{N+1} H^{(2)\top}$ and each $G_i^{(2)} D_i G_i^{(2)\top}$ is a diagonal matrix with positive diagonal entries. We use $B_3$ to denote the first matrix in the right hand side of (14) and $B_4$ to denote the second. In addition, other blocks of $\bar{A} D \bar{A}^\top$ are

$$\bar{A}_3 D \bar{A}_1^\top = 0,$$

$$\bar{A}_3 D \bar{A}_2^\top = \begin{bmatrix} \mathbf{1}_m^\top D_{N+1} H^{(2)\top} & \cdots & \mathbf{1}_m^\top D_{N+1} H^{(2)\top} \end{bmatrix} := \boldsymbol{\alpha}^\top,$$

$$\bar{A}_3 D \bar{A}_3^\top = \mathbf{1}_m^\top D_{N+1} \mathbf{1}_m := c.$$

With the new notations, we have

$$\bar{A} D \bar{A}^T = \begin{bmatrix} B_1 & B_2 & \mathbf{0} \\ B_2^\top & B_3 + B_4 & \boldsymbol{\alpha} \\ \mathbf{0} & \boldsymbol{\alpha}^\top & c \end{bmatrix}.$$

$\square$

## C   Proof of lemma 4.2

To justify lemma 4.2, we need the following basic result which can be verified through direct computation.

**Lemma C.1** *All the non-zero entries of matrices $B_1, B_2, B_3$ and $B_4$ are positive, and*

*a)* $B_3 \mathbf{1}_{M_2} = B_2^\top \mathbf{1}_M$.    *b)* $B_1 \mathbf{1}_M - B_2 \mathbf{1}_{M_2} > 0$.

*Proof.* a)

$$B_3 \mathbf{1}_{M_2} = \begin{bmatrix} G_1^{(2)} D_1 G_1^{(2)\top} \mathbf{1}_{m-1} \\ \vdots \\ G_N^{(2)} D_N G_N^{(2)\top} \mathbf{1}_{m-1} \end{bmatrix}, \quad B_2^\top \mathbf{1}_M = \begin{bmatrix} G_1^{(2)} D_1 F_1^\top \mathbf{1}_{m_1} \\ \vdots \\ G_N^{(2)} D_N F_N^\top \mathbf{1}_{m_N} \end{bmatrix}$$

Recall that $G_i^{(2)} = \mathbf{1}_{m_i}^\top \otimes [\mathbf{0}_{m-1}, I_{m-1}]$, $F_i = I_{m_i} \otimes \mathbf{1}_m^\top$, and $D_i$'s are diagonal matrices, we have $G_i^{(2)} D_i F_i^\top \mathbf{1}_{m_i} = G_i^{(2)} D_i G_i^{(2)\top} \mathbf{1}_{m-1}$ and thus $B_3 \mathbf{1}_{M_2} = B_2^\top \mathbf{1}_M$.

b)

$$B_1 \mathbf{1}_M = \begin{bmatrix} F_1 D_1 F_1^\top \mathbf{1}_{m_1} \\ \vdots \\ F_N D_N F_N^\top \mathbf{1}_{m_N} \end{bmatrix}, \quad B_2 \mathbf{1}_{M_2} = \begin{bmatrix} F_1 D_1 G_1^{(2)\top} \mathbf{1}_{m-1} \\ \vdots \\ F_N D_N G_N^{(2)\top} \mathbf{1}_{m-1} \end{bmatrix}$$

It is easy to verify that $F_i D_i F_i^\top \mathbf{1}_{m_i} > F_i D_i G_i^{(2)\top} \mathbf{1}_{m-1}$ and thus $B_1 \mathbf{1}_M - B_2 \mathbf{1}_{M_2} > 0$.    $\square$

With this basic lemma at hand, we are able to prove lemma 4.2.

**proof of lemma 4.2:**

*Proof.*

**a)** It is easy to verify the block-diagonal structure of $A_1$, so we just need to prove the positive definiteness and the strict diagonal dominance. Assume $A_1$ is not positive definite and $-\lambda \leq 0$ is an eigenvalue of $A_1$, then $\lambda I_{M_2} + A_1$ is a singular matrix.

From the results in lemma C.1, we have

$$
\begin{aligned}
& (\lambda I_{M_2} + A_1)\mathbf{1}_{M_2} \\
= \ & \lambda\mathbf{1}_{M_2} + B_3\mathbf{1}_{M_2} - (B_2^\top B_1^{-1}B_2)\mathbf{1}_{M_2} \\
= \ & \lambda\mathbf{1}_{M_2} + B_2^\top B_1^{-1}B_1\mathbf{1}_M - (B_2^\top B_1^{-1}B_2)\mathbf{1}_{M_2} \\
= \ & \lambda\mathbf{1}_{M_2} + B_2^\top B_1^{-1}(B_1\mathbf{1}_M - B_2\mathbf{1}_{M_2}) \\
> \ & \mathbf{0}_{M_2}
\end{aligned}
\tag{15}
$$

where the first equality is from a) of lemma C.1; the last inequality is from b) of lemma C.1 and the fact that $B_2^\top B_1^{-1} \geq 0$ and each row of $B_2^\top B_1^{-1}$ has at least one strict positive entry.

Since $B_1, B_2, B_3 \geq 0$, $B_3$ is a diagonal matrix, together with (15), we know that the diagonal entries of $\lambda I_{M_2} + A_1 = \lambda I_{M_2} + B_3 - B_2^\top B_1^{-1}B_2$ are positive and the off-diagonal entries are non-positive. Let $E_{M_2} := \mathbf{1}_{M_2}\mathbf{1}_{M_2}^\top - I_{M_2}$, then

$$
I_{M_2} \circ |A_1 + \lambda I_{M_2}| = I_{M_2} \circ (A_1 + \lambda I_{M_2}), \quad E_{M_2} \circ |A_1 + \lambda I_{M_2}| = -E_{M_2} \circ (A_1 + \lambda I_{M_2}),
$$

and

$$
(I_{M_2} \circ |A_1 + \lambda I_{M_2}|)\,\mathbf{1}_{M_2} - (E_{M_2} \circ |A_1 + \lambda I_{M_2}|)\,\mathbf{1}_{M_2} = (\lambda I_{M_2} + A_1)\mathbf{1}_{M_2} > \mathbf{0}_{M_2}
$$

This means $\lambda I_{M_2} + A_1$ is strictly diagonal dominant and thus nonsingular, which is a contradiction. Therefore, $A_1$ is positive definite. Take $\lambda = 0$ in the preceding analysis, we know $A_1$ is strictly diagonal dominant.

**b)** It is easy to verify that $A_2 = (\mathbf{1}_N\mathbf{1}_N^\top) \otimes (diag(\boldsymbol{y}) - \frac{1}{c}\boldsymbol{y}\boldsymbol{y}^\top)$. In view of the definition of $c$, we have $c > \mathbf{1}_{m-1}^\top\boldsymbol{y}$. Thus, the second claim of b) is a special case of a) with $B_1 = c$, $B_2 = \boldsymbol{y}^\top$ and $B_3 = diag(\boldsymbol{y})$.

$\square$

# D    Analysis of algorithm 2

In this section, we prove that through the steps in Algorithm 2, we get the accurate solution of the system $(A_1 + A_2)\boldsymbol{x} = \boldsymbol{g}$. We need a basic lemma on the inverse matrix on the sum of tow matrices.

**Lemma D.1** *Let $A \in \mathbb{R}^{n \times n}$ be an nonsingular matrix and $B \in \mathbb{R}^{n \times d}$, where $n$ and $d$ are two positive integers. Then*

$$
(A + BB^\top)^{-1} = A^{-1} - A^{-1}B(I_n + B^\top A^{-1}B)^{-1}B^T A^{-1}
$$

Recall that we have proved in lemma 4.2 that $Y$ is positive definite. Suppose $Y = R^\top R, R \in \mathbb{R}^{(m-1)\times(m-1)}$ and let $\tilde{R} = \mathbf{1}_N \otimes R^\top$. Then, $A_2 = \tilde{R}\tilde{R}^\top$. Further more, let

$$
\bar{R} = \begin{bmatrix} 0 \\ \vdots \\ 0 \\ 1 \end{bmatrix}_{N \times 1} \otimes R^\top, \ and \ U = \begin{bmatrix} I_{m-1} & & & I_{m-1} \\ & I_{m-1} & & \vdots \\ & & \ddots & I_{m-1} \\ & & & I_{m-1} \end{bmatrix}_{M_2 \times M_2}
$$

Note that $U$ is the same as defined in the main text part above Algorithm 2. It is easy to verify that $\tilde{R} = U\bar{R}$ and with the help of lemma D.1, we have

$$
\begin{aligned}
(A_1 + A_2)^{-1} &= \left(A_1 + \tilde{R}\tilde{R}^\top\right)^{-1} \\
&= A_1^{-1} - A_1^{-1}\tilde{R}(I + \tilde{R}^\top A_1^{-1}\tilde{R})^{-1}\tilde{R}^\top A_1^{-1} \\
&= A_1^{-1} - A_1^{-1}U\bar{R}(I + \tilde{R}^\top A_1^{-1}\tilde{R})^{-1}\bar{R}^\top U^\top A_1^{-1}.
\end{aligned}
$$

Define

$$
W := \bar{R}(I + \tilde{R}^\top A_1^{-1}\tilde{R})^{-1}\bar{R}^\top = \begin{bmatrix} 0 \\ \vdots \\ 0 \\ R^\top \end{bmatrix} (I_{m-1} + \sum_{i=1}^N RA_{ii}R^\top)^{-1} \begin{bmatrix} 0 & \cdots & 0 & R \end{bmatrix} \tag{16}
$$

then

$$
(A_1 + A_2)^{-1} = A_1^{-1} - A_1^{-1}UWU^\top A_1^{-1} \tag{17}
$$

From (16), it is clear that all entries of $W$ are zero, except for the last $(m-1) \times (m-1)$ block $W_{NN} = R^\top(I_{m-1} + \sum_{i=1}^N RA_{ii}R^\top)^{-1}R$. With further calculation,

$$
W_{NN} = \left(R^{-1}R^{-\top} + \sum_{i=1}^N A_{ii}^{-1}\right)^{-1} = \left(Y^{-1} + \sum_{i=1}^N A_{ii}^{-1}\right)^{-1}.
$$

To solve the system $(A_1 + A_2)\boldsymbol{x} = \boldsymbol{g}$ with the equation (17), we just need to let each term in (17) act on the vector step by step. That's exactly what Algorithm 2 does.

## E   Proof of theorem 4.3

In this section, we present the detailed analysis of computational cost and memory usage of SLRM and DLRM. Here we restate theorem 4.3.

**Theorem E.1**  *1). For SLRM, Algorithm 1, the time complexity in terms of flops is $O(m^2 \sum_{i=1}^N m_i + Nm^3)$, and the memory usage in terms of doubles is $O(m \sum_{i=1}^N m_i + Nm^2)$. 2). For the DLRM, the time complexity in terms of flops is $O(m \sum_{i=1}^N m_i^2 + \sum_{i=1}^N m_i^3)$, and the memory usage in terms of doubles is $O(m \sum_{i=1}^N m_i + \sum_{i=1}^N m_i^2)$.*

*Proof.* (1) First, for SLRM, assuming taking full advantage of the sparse structure, we count the flops required for computing each of the following quantities in Algorithm 1:

$$
B_1 : O(m \sum_{t=1}^N m_t); \ B_2 : 0; \ B_3 : O(m \sum_{t=1}^N m_t); \ T : O(m \sum_{t=1}^N m_t); \ A_1 : O(m^2 \sum_{t=1}^N m_t); \ A_2 : O(m^2);
$$

$$
\boldsymbol{z}^{(1)} : O(m \sum_{t=1}^N m_t); \ \boldsymbol{z}^{(2)} : O(Nm); \ \boldsymbol{z}^{(3)} : O(Nm^3); \ \boldsymbol{z}^{(4)} : O(Nm); \ \boldsymbol{z}^{(5)} : O(m \sum_{t=1}^N m_t).
$$

The computation of $A_1$ and $\boldsymbol{z}^{(3)}$ requires most flops. The total flops required for SLRM is $O(m^2 \sum_{t=1}^N m_t + Nm^3)$.

On the other hand, for implementation of the whole interior-point methods, the major data that should be kept in the memory include:

(a) Several vectors that is at the same level as a primal variable or a dual variable. Note that the scale of a primal variable is $m(\sum_{i=1}^N m_i) + m$ flops, and the scale of a dual variable is $\sum_{i=1}^N m_i + N(m-1) + 1$ flops.

(b) Matrix $\bar{A}$ which is defined in lemma 3.1. Recall that

$$\bar{A} = \begin{bmatrix} F_1 & & & & & & \\ & F_2 & & & & & \\ & & \ddots & & & & \\ & & & F_N & & & \\ G_1^{(2)} & & & & & & H^{(2)} \\ & G_2^{(2)} & & & & & H^{(2)} \\ & & \ddots & & & & \vdots \\ & & & & G_N^{(2)} & & H^{(2)} \\ & & & & & & \mathbf{1}_m^\top \end{bmatrix}$$

Since each column of $F_i$ and each column of $G^{(i)}$ has at most one non-zero element, the total number of non-zero elements in $F_1, \ldots, F_N$ and $G_1^{(2)}, \ldots, G_N^{(2)}$ is bounded by $2m\sum_{i=1}^N m_i$. In addition, $H^{(2)}$ has $m-1$ non-zero elements, so the total number of non-zero elements in $\bar{A}$ is bounded by $2m\sum_{i=1}^N m_i + N(m-1) + m$

(c) Diagonals of matrices $B_1$ and $B_3$, and diagonal blocks of matrices $B_2$ and $A_1$. The data scale of the diagonals of matrices $B_1$ and $B_3$ are even smaller than a dual variable. The diagonal blocks of matrices $B_2$ and $A_1$ have $m\sum_{i=1}^N m_i$ elements and $N(m-1)^2$ elements, respectively.

(d) Other intermediate vectors or matrices, whose data scale is bounded by a constant time of the data scale in (a), (b) and (c).

With the analysis in (a) to (d), we know the memory usage of SLRM is bounded by $O(m\sum_{i=1}^N m_i + Nm^2)$.

(2) The major difference of DLRM and SLRM is that, we don't need to formulate the diagonal blocks $\{A_{ii} : i = 1\ldots, N\}$ of matrix $A_1$ explicitly and compute the inverses of $A_{ii}$s. Instead, we need to compute $(B_{1i} - B_{2i}B_{3i}^{-1}B_{2i}^\top)^{-1}$ explicitly, which requires $O(m\sum_{i=1}^N m_i^2)$ flops for matrix multiplication and $O(\sum_{i=1}^N m_i^3)$ flops for matrix inverse. Since all other matrix-vector operations are cheap compared with matrix multiplication and inverse, as a result, the leading cost of the computation time is at the level $O(m\sum_{i=1}^N m_i^2 + \sum_{i=1}^N m_i^3)$.

Further more, since we need to keep $(B_{1i} - B_{2i}B_{3i}^{-1}B_{2i}^\top)^{-1}, i = 1, \ldots, N$ in memory instead of $A_{ii}^{-1}$, with simply different analysis as in part(1), we know the memory usage of DLRM is at the level $O(m\sum_{i=1}^N m_i + \sum_{i=1}^N m_i^2)$.

$\square$

# F  Examples of local minima and saddle points in free support cases

**An example of local minima**

Set $\Pi^{(t)} = \left[\boldsymbol{\pi}_1^{(t)\top}, \boldsymbol{\pi}_2^{(t)\top}, \ldots, \boldsymbol{\pi}_m^{(t)\top}\right]^\top$. Let $N$ be any positive integer and $m = 2$, $d = 1$, $m_t = 3$, $Q^{(t)} = [0, 0.9, 1.1]$ and $\boldsymbol{a}^t = [0.01, 0.495, 0.495]$. Then $X = [0, 1]$, $\boldsymbol{w} = (0.01, 0.99)$ and $\boldsymbol{\pi}_1^{(t)} = (0.01, 0, 0)$ and $\boldsymbol{\pi}_2^{(t)} = (0, 0.495, 0.495)$ is a local minimum. But it is not a global minimum because a lower objective value occurs when $X = \{0.9, 1.1\}$, $\boldsymbol{w} = (0.505, 0.495)$, $\boldsymbol{\pi}_1^{(t)} = (0.01, 0.495, 0)$ and $\boldsymbol{\pi}_2^{(t)} = (0, 0, 0.495)$.

**An example of saddle point**

Let $N$ be any positive integer and $m = 2$, $d = 1$, $m_t = 3$, $Q^{(t)} = [0, 1/2, 3/2]$ and $\boldsymbol{a}^t = [1/3, 1/3, 1/3]$, then $X = [0, 1]$, $\boldsymbol{w} = (1/3, 2/3)$, $\boldsymbol{\pi}_1^{(t)} = (1/3, 0, 0)$ and $\boldsymbol{\pi}_2^{(t)} = (0, 1/3, 1/3)$ is a saddle point. Fixing $X$, the $\boldsymbol{w}$ and $\Pi^{(t)}$ is an optimal basic solution of problem 4. Fixing $\boldsymbol{w}$ and $\Pi^{(t)}$, $X$ is the solution of (7). It is not a local minimum, because a lower objective value of problem 4 can occur when $X = \{\delta, 1\}, \forall \, \delta \in (0, 1/2)$.

# G  Details of MAAIPM

Figure 7 visualizes the primal variables $x_i$ and objective gradients $c_i$ in each iteration of MAAIPM.

Figure 7: The primal variables and objective gradients in different iterations of MAAIPM. $x_i$ is returned by each iteration of IPM under the objective gradient $c_i$, and $c_i$, $i = 5, \ldots, 11$, is calculated by $x_{i-1}$ according to (8). At the beginning, MAAIPM updates objective gradient after every a few primal-dual IPM iterations(green). Then MAAIPM applies primal IPM(yellow and red) to frequently update objective gradient $c$ and uses "jump" tricks to escape local minima. $x_6$ and $x_9$ are the first primal variables returned by one primal IPM iteration form a smartly chosen starting point.

---

**Algorithm 3:** Matrix-based Adaptive Alternating Interior-point Method(MAAIPM)

---

**Input:** an initial $X^0$

1 **if** *support points are pre-specfied* **then**
2      implement predictor-corrector IPM;
3      **Output** $\boldsymbol{w}^*, \{\Pi^{(t),*}\}$

4                                                      ▷ Pre-specified support cases
5 **while** *at the beginning* **do**
6      predictor-corrector IPM to solve (4) and update $X^*$;

7                              ▷ Update support $X^*$ every a few IPM iterations
8 **while** *a termination criterion is not met* **do**
9      s = 0, apply the warm-start strategy to smartly choose the starting point;
10      **while** *the penalty $\mu^s$ is not sufficiently close to 0* **do**
11          calculate the Newton direction $p^s$ at $(\boldsymbol{w}^s, \{\Pi^{(t),s}\})$ by (11);
12          $(\boldsymbol{w}^{s+1}, \{\Pi^{(t),s+1}\}) = (\boldsymbol{w}^s, \{\Pi^{(t),s}\}) + \alpha^s p^s$, where $\alpha^s$ ensures the interior point;
13          update $X^*$ by (7) and choose penalty $\mu^{s+1} < \mu^s$;
14                           ▷ Update support $X^*$ every IPM iteration
15          s = s + 1;

16                                                      ▷ "Jump" tricks
**Output:** $\boldsymbol{w}^s, X^*, \{\Pi^{(t),s}\}$

---