[Reviews · NeurIPS 2019]

Reviewer 1



Review -------- This paper is very well written and easy to follow. The Wasserstein barycenter problem computes the Frechet mean of a set of probability distributions. When the support of the unknown barycenter is fixed, the problem is a LP problem. If not, one can alternatively minimize with respect to the sample points and the probability weights. The authors propose a general accelerated algorithm for both cases with a cubic complexity with respect to the dimension. The main finding of this work is that their method solves the OT problem without strictly convex regularization (such as entropy) to speed up the computation. Thus, they manage to obtain more accurate estimations of the barycenter in a competitive time. Questions --------- 1. In L.79-80, the authors mention "Our algorithms also inherits 80 a natural structure potentially fitting parallel computing scheme well". What does "potentially" exactly mean ? Since Sinkhorn's algorithm (IBP) is parallelizable on the N axis (across number of input distributions), this statement deserves to be discussed. 2. Perhaps I am missing something but if the complexity of the proposed method is cubic w.r.t to the dimension, how come the plots of figure 4 show a linear rate on the second row ? When varying m, does the dimension of ALL distributions not change at the same time ? 3. the second column of Figure 4 is not clear. The computation time is not already displayed on the first column ? And why not all methods are displayed on the last 3 columns ? 4. On regular grids such as images, the iterations of Sinkhorn's algorithm can be written as Kernel convolutions (if the squared l2 distance is used as a ground metric, which is the case here). This idea was introduced in https://people.csail.mit.edu/jsolomon/assets/convolutional_w2.compressed.pdf but is not very clear in that paper. Briefly, each iteration can be seen as a Gaussian smoothing which -- thanks to the separability of the squared l2 -- can be applied on rows and columns independently. This results in a reduced complexity O(Nm^2) => O(Nm^3/2) of each iteration of N 2D distributions in R^m. This trick is crucial and should have been used when comparing with Sinkhorn.

Reviewer 2



================================================== AFTER AUTHOR FEEDBACK. I maintain my score. However, I'm not convinced by the author's response to my question on showing the derivation of the previous best run time. They need to use more current algorithms for LP (for example, the one by Cohen, Lee, and Song) in order to compute how fast this problem can be solved. Without a derivation using these algorithms, it's unclear *how much* of an improvement this submission provides. Without a clear picture of this, it's unclear to determine the full contribution of the paper. ================================================== ORIGINALITY See comment in 1. The paper recognizes and exploits some properties of the objects in the problem in a way that lets the authors improve the algorithm's runtime by orders of magnitude. This is an original observation/contribution. However, I am concerned about the reported run time for the previous best interior point algorithm: in particular, in the supplementary material provided, in line 69 (paragraph titled "Low Theoretical Complexity"), the previous best run time is reported to be O(N^3 m^4), but there is no citation provided or computation shown for this claim. In order for me to be convinced that the authors' work does have a significant improvement upon previous work, I'd like to request the following from them. (1) an explicit computation showing how the previous best run time of O(N^3 m^4) is computed, (2) applying the run times from some classical and contemporary results for LPs: papers by Vaidya https://link.springer.com/article/10.1007/BF01580859, Lee-Sidford https://arxiv.org/abs/1312.6713, and current best LP result by Cohen-Lee-Song https://arxiv.org/abs/1905.04447. QUALITY - Overall, I found the paper to be high quality, with adequate citations for various results. However, I do have the following concerns: I am confused why the NeurIPS paper by Cuturi is cited for Iterative Bregman Projection (line 249 in the supplementary pdf) when that paper doesn't even mention Bregman projections. For Figure 2, I think the evaluation of SLRM and DLRM must be done against more recent solvers for normal equations such as those mentioned in the "Originality" paragraph above and also other interior point solvers which are more specialized (than the MATLAB solver which I'm not sure exploits the structure of normal equations). CLARITY - Overall, the paper reads quite well. However, I think the clarity can be improved in the following places: In line 115, the authors describe an LP and then all the vectors/variables/matrices in it; however, the vector c is not defined. Also, it would help to clearly state, separately, the problem dimension and number of constraints at this point. It would also greatly help if you gave the reader a "sense" of each of the m_i's, so that the reader can get a better sense of m sum_i m_i + m and Nm + sum_i m_i + 1 (just as you have done in the section "Low Theoretical Complexity" in the supplementary material, where you say, let m_i = m). In Algorithm 1, it would help to have caption or text saying that z(i) is the solution to which system; the way it's written now, I found it difficult to understand from the procedure alone that it's actually solving (bar_A D bar_A^t ) z= f. Finally, the authors are using the cost of an n x n matrix inversion to be n^3, when this should be n^\omega. The current value of \omega is roughly 2.372927 (see Table 3 by Le Gall and Urrutia https://arxiv.org/abs/1708.05622), so I think this should be made more clear. SIGNIFICANCE: I think the formulation and use of the block structure is certainly novel for this problem, and anyone who tries to solve this problem can benefit from the authors' observations. However, as someone who has not worked on the problem of Wasserstein Barycenter before, I don't think I can judge the importance of the problem itself. I think the ideas of using structure could be used in other interior point methods for LPs.

Reviewer 3



-------- After Author Feedback --------- I stand by my initial review and score. I believe the paper presents an interesting new approach to computing Wasserstein barycenters that is well motivated, and well tested. My initial misgivings were addressed sufficiently in the rebuttal. ----------------------------------------------- Typo: 93: Wasserstein distance instead of barycenter The paragraphs from 132--148 imply that the algorithm recovers a globally optimal solution to the free support problem. Can you elaborate or clarify this? A globally optimal solution to the Wasserstein barycenter problem implies a globally optimal solution to k-means which is known to be NP-hard. In lines 237--239 it is mentioned that the memory usage for MAAIPM is similar to that of Sinkhorn type algorithms. The memory usage in Sinkhorn algorithms is dominated by the cost of storing the kernel matrix K = exp(-d(x, y) / epsilon). However, there are work-arounds for this issue that only require applying an operator to a vector, and thus reduce memory requirements to O(n + m). Have the authors compared to these solutions, for example: Solomon et al. "Convolutional Wasserstein Distances: Efficient Optimal Transportation on Geometric Domains." In addition to Gurobi, can the authors compare to CPLEX and Mosek. I have frequently found those to be faster than Gurobi for transport problems. Another point of comparison that is also tailor made for transport problems can be found here: https://github.com/nbonneel/network_simplex (it's true that this is for computing distances, but would be a good point of comparison nonetheless). The authors can also compare with their reference [17] for a regularized transport solution with faster convergence than Sinkhorn, with reference [2] for a empirically faster Sinkhorn iteration scheme with better memory requirements, or with - Altschuler et al., "Massively scalable Sinkhorn distances via the Nyström method." for a very fast and memory efficient approximation of Sinkhorn. I hope the authors provide code later on. In brief: Originality: A novel algorithm for computing unregularized Wasserstein barycenters via a different interior point method to solve the associated linear program. Quality: The paper is technically sound, and the claims are well supported by thorough experiments. The algorithms are described in full and could be easily implemented independently. Clarity: The paper is readable, the algorithms explained well, and the experiments documented clearly. Significance: One shortcoming of this paper is lack of comparison with newer, faster versions of regularized optimal transport (mentioned above). Unregularized optimal transport is of independent interest, but this submission would be strengthened if MAAIPM is faster than e.g. Greenkhorn.

[Author Response · NeurIPS 2019]

We deeply thank reviewers for their insightful and constructive comments, which greatly helped us improve the work. As all 3 reviewers pointed out, **the main novelty and contribution of this work are that we found that an adapted interior point algorithm can be much more efficient than traditionally expected both theoretically and computationally in solving the Wasserstein Barycenter problem. The creativity lies in reducing the complexity of solving the normal equations in each iteration by fully exploiting its matrix's block diagonal structure. This finding is potentially useful in further speeding up many state-of-art interior point algorithms that mainly focus on reducing the iteration numbers.** Compared to popular regularization approaches such as Sinkhorn-based methods, this algorithm may gain a good balance between accuracy and speed.

Some common issues from reviewers are addressed here: we conducted more experiments to compare with Solomon et al.'s convolutional Sinkhorn algorithm(will keep trying more experiments!), and the comparison result is similar to that of IBP. For reproducibility, codes are on GitLab now. We will post the link in the revision.

**Response to comments of Reviewer 1:** Thank you for very insightful inputs!

Q1: Our methods for solving normal equations is parallelizable on the N axis, too. In SLRM/DLRM, the major cost is computing $A_{ii}^{-1}$ for each $i = 1, \ldots, N$ independently(in Algorithm 2 line 1), which can be done in parallel for all $i$'s.

Q2: It's a crucial observation! Thanks for raising it. In Thm. 4.3: the complexity of one iteration in our IPM is linear w.r.t. $N$ and $m$ (or $m_i$). This is because we directly build up $\bar{A}D\bar{A}^\top$ and the complexity of one IPM iteration is the same as the complexity of SLRM/DLRM. We'll explain it more in the paper. Thus, in figure 4, if we fix 2 parameters among $N$, $m$, $m_i$ and change the other one, it'll show a linear rate. When varying $m$, the pattern is similar to figure 4.

Q3: We use 2 columns to demonstrate the linear relation between runtime and N(m'). We regard Gurobi's solution accurate and omit it in the middle 2 columns. In the right column, we compare with Gurobi to display MAAIPM's low memory occupancy advantages over traditional second order methods, though it does take more memory than first order methods by nature. It is great advice to display all methods. We'll add it in the paper.

Q4: Thanks for this important advice! We have tested the method and it indeed improved the results from Sinkhorn's side! We will add it in the paper.

Improvement suggestions: Thanks for detailed instructions! We have followed each point carefully and revised the paper accordingly and will include them in the revision.

**Response to Reviewer 2:** Thanks! Your concerns helped us strengthen some important issues of the paper!

Q1: "However, I am concerned...run time for the previous best interior point algorithm... reported to be $O(N^3m^4)$..."

ANS: Thanks for raising up this issue! It's the most important theoretical part and we should have explained it better(will revise!). The paper focuses on reducing the complexity of solving the normal equation in one iteration. For a standard LP, let $\bar{m}$, $\bar{n}$ be the number of constraints and variables. Then computing $ADA^T$ requires $O(\bar{m}^2\bar{n})$ flops, while solving a system with the coefficient matrix $ADA^T$ requires $O(\bar{m}^3)$ flops. So a single IPM iteration takes $O(\bar{m}^2\bar{n})$. If $m = m_i$, the LP has $Nm^2 + m$ variables and $2Nm + 1$ constraints. Thus $O(N^3m^4)$ directly comes from formulating and solving the normal equations in one iteration via direct matrix multiplication and Cholesky decomposition. Possibly $O(N^3m^4)$ can be improved by techniques such as fast matrix multiplication, but not too much. We'll add more related results(especially overall complexity) and look into the valuable references you provide.

Q2: "I am confused why the NeurIPS paper by Cuturi is cited for Iterative Bregman Projection......"

ANS: Thank you for pointing this out! It is a mistake and we'll correct it.

Q3: "I think the evaluation of SLRM and DLRM must be done against more recent solvers for normal equations......"

ANS: We totally agree! Though the computing time of normal equations in a single iteration can't be exacted accurately from commercial solvers(told by solver developers), we'll try our best to figure out.

Q4: " I think it doesn't make much sense to compare an interior point method with first order methods......"

ANS: Thanks for very meaningful comments! We did compare our algorithm with best commercial solvers using interior points methods(Gurobi/MOSEK, and CPLEX later). We also compared with mainstream methods in this area– even though they are totally different types of methods– to show its competitiveness.

The comments on improving the clarity and the fast matrix inversion are greatly appreciated. They can improve the theoretical complexity of our SLRM/DLRM. The revision will be made accordingly!

**Response to Reviewer 3:** Thank you for sharp observation and kind advice!

Q1: "The paragraphs from 132–148 ..." ANS: Thanks for pointing out the problem! Our writing here isn't clear. MAAIPM cannot optimally solve the free support case. We'll rewrite this more carefully.

Q2: "In lines 237–239 ..." ANS: Thanks for great suggestion! We have finished the experiment and will add it in the paper. The result shows that our memory usage is higher than popular first order methods, but it's usually within constant times of them. Essentially our advantage lies in the balance of accuracy and speed.

Q3: "In addition to Gurobi ..." ANS: We thank the referee for raising an excellent point. Mosek was tested and is much faster than Gurobi, though it is still roughly 3 times slower than MAAIPM and occupies more memory than MAAIPM for large problems. We will include it in the paper and will test CPLEX later. MAAIPM possesses not only computation speed advantage, but also computation memory efficiency.

Q4: "The authors can also compare ..." ANS: Thanks for helpful suggestions! We've been testing newer algorithms and will add those in the revision besides Solomon et al.'s convolutional Sinkhorn algorithm.

[Meta-Review · NeurIPS 2019]

The reviewers converged to a consensus of accepting this paper! Congratulations. In your revision, please address all comments in the reviews and include promised experiments mentioned in the rebuttal. One condition for us to accept your revision: Please add a clear derivation of the previous best run times for this problem. Also revise in light of the fact that the current method of computing the previous best run time (as they described in the rebuttal) is not quite correct, since it does not use the current fastest LP algorithms for the computations (naive methods have issues that were circumvented by the papers mentioned in the review).